# A fluorogenic, peptide-based probe for the detection of Cathepsin D in macrophages

Maria Rodriguez-Rios [1], Brian J. McHugh[2], Zhengqi Liang[1], Alicia Megia-Fernandez [1,3], Annamaria Lilienkampf[1], David Dockrell[2] & Mark Bradley [4✉]

Cathepsin D is a protease that is an effector in the immune response of macrophages, yet to date, only a limited number of probes have been developed for its detection. Herein, we report a water soluble, highly sensitive, pH insensitive fluorescent probe for the detection of Cathepsin D activity that provides a strong OFF/ON signal upon activation and with bright emission at 515 nm. The probe was synthesised using a combination of solid and solution-phase chemistries, with probe optimisation to increase its water solubility and activation kinetics by addition of a long PEG chain (5 kDa) at the C-terminus. A BODIPY fluorophore allowed detection of Cathepsin D across a wide pH range, important as the protease is active both at the low pH found in lysosomes and also in higher pH phagolysosomes, and in the cytosol. The probe was successfully used to detect Cathepsin D activity in macrophages challenged by exposure to bacteria.

[1] School of Chemistry, University of Edinburgh, David Brewster Road, EH9 3FJ Edinburgh, UK. [2] University of Edinburgh Centre for Inflammation Research, Queen's Medical Research Institute, 47 Little France Crescent, Edinburgh BioQuarter, Edinburgh EH16 4TJ, UK. [3] Organic Chemistry Department, Faculty of Sciences, University of Granada, Avda. Fuente Nueva S/N, Granada 18071, Spain. [4] Precision Healthcare University Research Institute, Queen Mary University of London, Empire House, 67-75 New Road, London E1 1HH, UK. ✉email: m.bradley@qmul.ac.uk

Macrophages are tissue resident immune cells that are essential for fighting infection and the maintenance of tissue homoeostasis[1]. They are important during the initial stages of combating a bacterial infection, as well as resolution of the inflammatory response, and for bacterial clearance[2]. However, the intracellular killing and clearance capacity of macrophages can become "exhausted", allowing certain bacteria to develop survival strategies that evade elimination[3]. Recently, a mechanism known as apoptosis-associated bacterial killing has been proposed to enhance bacterial eradication. This process has been extensively studied in *S. pneumonae* infection models, whereby upon exhaustion of the intracellular killing processes, apoptosis of the macrophage is triggered *via* a multi-step pathway involving Cathepsin D (CatD). This aspartic protease is usually confined to the lysosome, hence why it requires an acidic environment to be optimally functional (optimal pH is 4.0)[4,5]. However, in the phagosomes of macrophages, the pH becomes more basic and from here, the protease can be released to the cytoplasm where the pH is approximately 7.4. The analysis and detection of CatD activity across the different cellular compartments would provide a tool to better understand the bacterial killing processes mediated by macrophages and apoptosis-associated bacterial killing.

Current CatD detection methods include the use of fluorescently labelled antibodies and BODIPY-labelled irreversible inhibitors based on Pepstatin A[6,7]. However, such probes offer no signal amplification and fluorescence is "always on" and, as such, suffer from poor signal-to-noise ratios[8]. In addition, Pepstatin A is a generic aspartic acid protease inhibitor, while antibodies cannot distinguish between active and non-active forms of the protease[9]. Efficient chromogenic substrates have long been reported, for example using the amino acid *p*-nitrophenylalanine[10], while a limited number of probes have been reported for CatD analysis, that include a dual MRI contrast probe decorated with the pH sensitive dye Oregon Green with cleavage removing an attached cell delivery peptide[11]. Fluorogenic, substrate-based CatD probes would offer enhanced signal-to-noise ratios due to amplification of the fluorescence signal, with thousands of copies of the probe being activated by a single enzyme molecule. FRET-based fluorogenic substrates for CatD have been reported, but typically in the blue region (350–495 nm) and often with fluorophores that show pH sensitivity that limits their application[1,3,12,13]. This is important as CatD is stored and active in acidic lysomes (pH 4–5), but can also mediate lysosomal effects by fusion with other organelles such as phagosomes where the pH is higher[14], and can also be released into the cytosol where it retains some residual activity. Therefore, it is important that any fluorophore for such probes is chosen such that the fluorescence signal is not influenced by pH. Fluorescein and its derivatives are widely used fluorophores and, although highly fluorescent at physiological pH, their fluorescence decreases with decreasing pH[15]. Rhodamine B, another xanthene-based fluorophore, is also pH dependent but with fluorescence quantum yields that increase with decreasing pH[16]. Although such pH dependency can be useful when developing probes that act as pH sensors[17], it is less useful when detecting biomarkers that might be present in different cellular compartments or involved in biological processes where pH changes manifest, as it becomes impossible to distinguish between enzymatic activity levels and simple pH variation. In such situations, it is vital that pH insensitive fluorophores are used, such as those based on the BODIPY or Cyanine scaffolds. A Cy3/5-based FRET substrate for CatD has been reported[18], but this was conjugated to a 20-mer oligonucleotide and used a substrate that showed poor cleavage kinetics. In addition, DNA oligomer attachment to a protease substrate poses a scale-up challenge and requires cell transfection for intracellular delivery.

Herein, we report the design and synthesis of a fluorogenic probe, based on a specific substrate for CatD, which provides an OFF→ON signal upon activation with good signal amplification. The substrate was originally derived by Pimenta et al.[19] from the natural CatD substrate kallistatin (Ala-Ile-Ala-Phe↑Phe-Ser-Arg-Gln), which has good affinity for the protease ($K_M = 0.27$ μM, $k_{cat} = 16.25$ s$^{-1}$). When choosing a substrate to build an optical probe to be used in cells and potentially in vivo, control of cross-reactivities is vital and it is important to design a specific substrate that is not cleaved by related proteases. One of the most abundant and active proteases in the lysosome is Cathepsin B, which prefers substrates bearing Phe-Arg or Arg-Arg sequences at the P2-P1 sites. Previously reported highly efficient cleavable substrates for CatD contain the sequence Phe-Arg (P1'-P2') which would also be cleaved by Cathepsin B. Thus, the substrate reported by Pimenta[19] was chosen for its optimal cleavage site combination, and its high catalytic efficiency, avoiding the sequences Phe-Arg and Arg-Arg, and adapted to be compatible with live cell imaging, by improving solubility and moving the emission of the fluorescence into the green region of the spectrum.

The new probe designed here was FRET-based, with a BODIPY fluorophore at the *N*-terminus, while the *C*-terminal Gln was replaced with a Lys residue functionalised with a quencher (Methyl Red, $\lambda_{max}$ 500 nm)[20]. The functionalised BODIPY derivative **2** ($\lambda_{ex/em}$ 503/516 nm) was selected due to its photostability and bright fluorescence ($\Phi = 0.6$)[21], while importantly being pH insensitive (unlike carboxyfluorescein[15]).

## Results and discussion

The first probe designed for CatD detection was **CatD-P1**, which was synthesised using a combination of solid and solution-phase synthesis (Figs. 1 and 2). In the design of a protease-based probe, eliminating cross-reactivity is important, with the need to choose a peptide substrate that is not cleaved by related proteases. Some of the previously reported substrates for CatD were derived from the inhibitor pepstatin A[22] or contained an Arg residue at the P2' position (Phe-Arg at P1'-P2')[13,23–26] that can lead to Cathepsin B cross-reactivity, but Pimenta et al. reported that a Ser residue at the P2' position that overcomes this problem[19].

The Methyl Red (MR) quencher was incorporated by coupling Fmoc-Lys(MR)-OH[20] onto a Rink-amide functionalised polystyrene resin (0.7 mmol/g), followed by the substrate sequence, using DIC and Oxyma as the coupling combination. The use of Fmoc-Lys(MR)-OH as a building block avoided the need for orthogonal deprotection on the resin[27,28]. A *bis*-ethylene glycol spacer was introduced as the last amino acid before the fluorophore to increase hydrophilicity and promote aqueous solubility of the probe. The BODIPY (1,3,5,7-tetramethyl-8-phenyl-4,4-difluoroboradiazaindacene) fluorophore chosen as part of the FRET pair is not stable to acids such as TFA or HCl[29], thus, the BODIPY-NHS ester **2**[30] was coupled in solution after the acidic cleavage from the resin, to give the desired FRET enabled peptide.

However, no significant increase in fluorescence was observed when **CatD-P1** was incubated with CatD (SI, Supplementary Fig. 1). This was attributed to the poor aqueous solubility of the probe, which resulted in precipitation during incubation (even when 10% DMSO was used) and led to the synthesis of probes **CatD-P2** and **CatD-P3** (Figs. 1 and 2). PEGylation of drugs and other biologically relevant molecules is a widely used approach to improve solubility[31] and here two PEGylated versions of the FRET probe were designed to enhance solubility, with the first **CatD-P2** synthesised by coupling six ethylene glycol moieties onto the C-terminus of the probe to give **CatD-P2** (Fig. 2). **CatD-P3** was synthesised by coupling Fmoc-Lys(N$_3$)-OH onto the Rink-amide linker (attached to a polystyrene resin) followed by

**Fig. 1 FRET-based Cathepsin D probes CatD-P1, CatD-P2, and CatD-P3.** The structures of the FRET-based Cathepsin D probes **CatD-P1, CatD-P2,** and **CatD-P3** incorporating a BODIPY fluorophore (highlighted in green, $\lambda_{ex/em}$ 503/516nm) at the *N*-terminus as the fluorescence donor and Methyl Red (highlighted in red, $\lambda_{max}$ 500 nm) as the quencher/acceptor. Cathepsin D cleaves the substrate between the two phenylalanine residues (shown in blue).

coupling the amino acids as described above. Following cleavage and BODIPY dye attachment, CuAAC[32,33] allowed the efficient attachment of a 5 kDa PEG-alkyne moiety (the reaction was monitored by HPLC, SI Supplementary Fig. 2). **CatD-P2** proved to be more water soluble than **CatD-P1**, but the addition of 3% DMSO was still required for enzymatic assays, with a maximum probe concentration of only 7–8 µM achievable in 1% DMSO/buffer. A 30-fold increase in fluorescence was seen with **CatD-P2** (20 µM) after incubation with CatD (SI, Supplementary Fig. 3). The probe **CatD-P3** with a 5 kDa PEG, however, was fully water soluble and was rapidly activated by CatD resulting in a 26-fold increase in fluorescence (Fig. 3a).

The presence of the large PEG unit had little effect on the enzymatic activity of **CatD-P3**, with a good binding affinity ($K_M$ of 8 µM) and reasonable turnover number ($k_{cat}$ of 0.8 s$^{-1}$ giving a catalytic efficiency ($k_{cat}/K_M$) of $7.5 \times 10^4$ M$^{-1}$s$^{-1}$ (SI, Supplementary Fig. 6), with the activation efficiently inhibited by pepstatin A (Fig. 3b). MALDI-TOF MS analysis of the enzymatic reaction confirmed that the probe was cleaved by CatD between the two Phe residues (SI, Supplementary Fig. 4). The fluorescence signal of the activated probe was not affected by pH (pH 4.0–7.4), showing that this water-soluble probe would allow detection of CatD in any relevant cellular compartment (SI, Supplementary Fig. 5). **CatD-P3** was photostable and functional in serum, i.e., in an environment with an abundance of proteins and other metabolites (Fig. 3c and SI Supplementary Fig. 9). **CatD-3** showed good specificity for CatD over other proteases present in macrophages, such as Cathepsins G and B, with other inflammatory proteases such as human neutrophil elastase (hNE) showing only low levels of cross-cleavage, although more cleavage was observed with Proteinase 3 (Fig. 3d)[34]. Cathepsin E, also showed some cleavage, which was expected due to both proteases sharing very similar substrate specificities[35,36].

**CatD-P3** was used for imaging CatD activity in human monocyte-derived macrophages (MDM). Macrophages were exposed to *S. pneumoniae* for 10 h to stimulate conditions inducing CatD activation[3], after which the macrophages were incubated with **CatD-P3** (10 µM) and imaged by confocal microscopy (Fig. 4). Control cells (unexposed to bacteria) showed low background fluorescence levels, while the fluorescence signal was enhanced in *S. pneumoniae* challenged macrophages due to activation of the probe. Since macrophages are professional phagocytes, we believe that probe uptake is via a combination of passive diffusion and phagocytosis. CatD has an optimal pH of 4.0, a pH found in the lysosomes and at this pH the cleaved probe is highly fluorescent. Thanks to its pH insensitivity (SI, Supplementary Fig. 5), during transport to or from the phagosome, for example when leaking into the cytoplasm (Fig. 4), the cleaved probe will remain fluorescent. Some of the activated probe was detected in the cytoplasm, potentially coming from the bacterial induced lysosomal leakage[37]. When the inhibitor Pepstatin A was present in the cell media, a decrease in the fluorescence signal was observed in the infected cells confirming that the increase in fluorescence resulted from probe activation (Fig. 4). LDH cyto-toxicity assays on macrophages incubated with 10 µM **CatD-P3** showed very low levels of toxicity at 1 h (0.27% +/−0.2%) and 6 h (6.3% +/−0.68%) (SI, Supplementary Fig. 7). Additionally, the cells appeared phenotypically normal after probe incubation by phase contrast microscopy (Fig. 4, brightfield images).

## Conclusions
A highly sensitive, pH insensitive FRET-based probe for CatD was synthesised, with conjugation of a 5 KDa PEG unit enabling excellent water solubility. The probe successfully entered cells and was able to detect CatD activity in human human monocyte-derived macrophages that were challenged by exposure to bacteria, while pretreatment of macrophages with inhibitor Pepstatin A resulted in a marked decrease in fluorescent signal. Current cellular imaging assays are focusing on evaluating the probe in more complex environments including as a useful tool to understand the role of CatD in apoptosis-associated bacterial killing in macrophages.

## Methods
All Fmoc-amino acids, DIC, Oxyma and Aminomethyl Polystyrene resin (0.7 mmol/g) and Fmoc-Rink Amide Linker were purchased from GL Biochem, Sigma, Fluorochem or Apollo

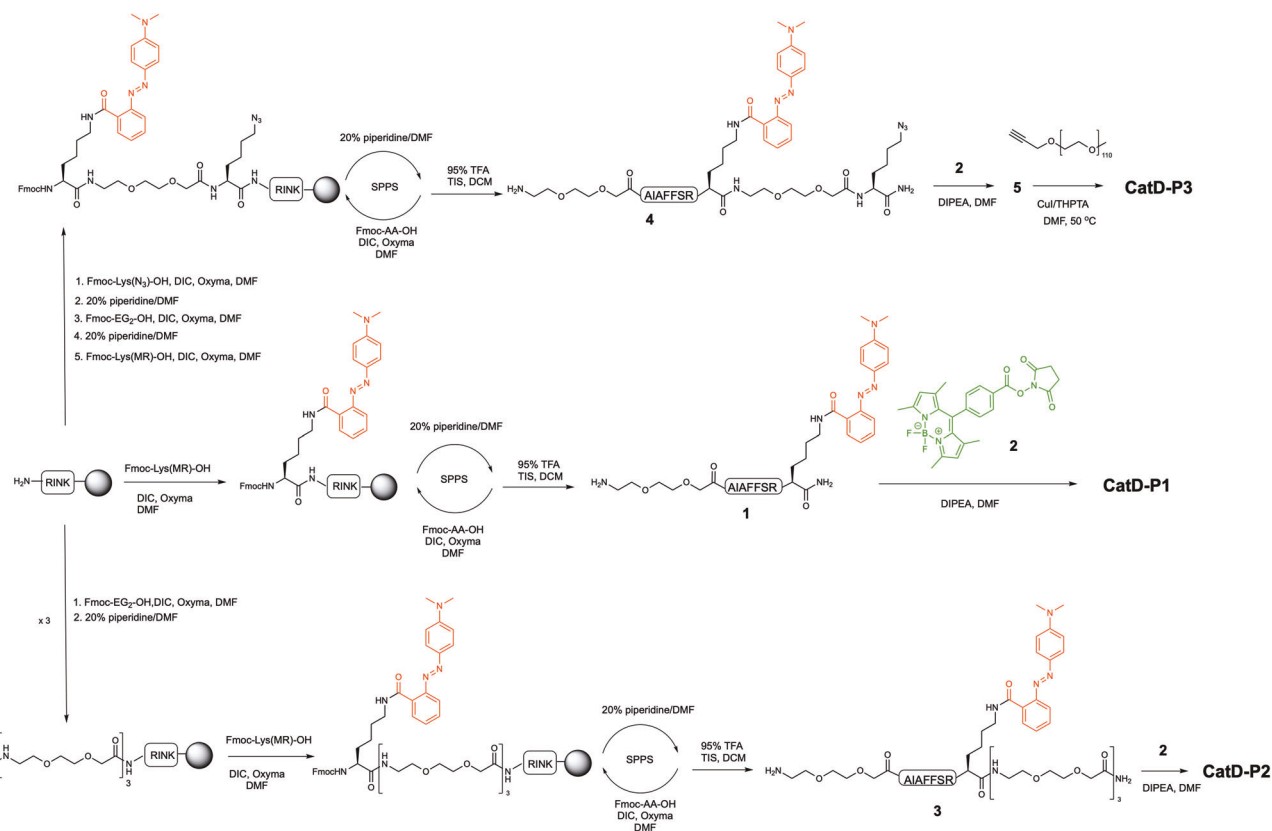

**Fig. 2 Synthesis of the pH insensitive CatD probes CatD-P1, CatD-P2 and CatD-P3.** All the probes were synthesised using a dual Fmoc/$^t$Bu solid-phase/solution-phase strategy. The peptides were fully deprotected and cleaved off the resin before incorporation of the BODIPY fluorophore onto the *N*-terminus as an activated-NHS ester. For **CatD-P1**, Fmoc-NH-(EG)$_2$-OH (*bis*-ethyleneglycol unit) was used as a spacer. Fmoc-Lys(N$_3$)-OH was incorporated as the first amino acid in the synthesis of **CatD-P3** for later conjugation of a 5 KDa polyethylene glycol (PEG) moiety via a copper-catalysed azide-alkyne cycloaddition (CuAAC).

Scientific and used without further purification. Analytical reverse-phase high-performance liquid chromatography (RP–HPLC) was performed on an Agilent1100 system equipped with a Kinetex 5 μm XB-C18 reverse-phase column (5 cm × 4.6 mm, 5 μm) with a flow rate of 1 mL/min where method A or B used a gradient eluting with H$_2$O/CH$_3$CN/HCOOH (95/5/0.1) to H$_2$O/CH$_3$CN/HCOOH (5/95/0.1) over 6 min or 10 min, respectively, holding at 95% CH$_3$CN for 3 min, with detection at 495 nm and by evaporative light scattering. NMR spectra were recorded on an automated Bruker AV500 in the indicated deuterated solvents at 298 K. Chemical shifts are reported on the δ scale in parts per million (ppm) and are referenced to the residual non-deuterated solvent peak for $^1$H NMR, and to the deuterated carbon of the solvent for $^{13}$C NMR. Coupling constants (*J*) are given in Hertz. Electrospray ionisation mass spectrometry (ESI–MS) analyses were carried out on an Agilent Technologies LC/MSD quadrupole 1100 series mass spectrometer (QMS) in an ESI mode. High-resolution mass spectra were recorded on a Bruker SolariX Fourier transform ion cyclotron resonance mass spectrometer (FT-MS). MALDI-TOF spectra were acquired on a Bruker Ultraflextreme MALDI TOF/TOF with a matrix solution of sinapic acid (10 mg/mL) in H$_2$O/CH$_3$CN/TFA (69.9/30/0.1) or α-cyano-4-hydroxycinnamic acid (10 mg/mL) in H$_2$O/CH$_3$CN/TFA (49.9/50/0.1). BODIPY–NHS[30] **2** and Fmoc-Lys(MR)-OH[20] were synthesised as previously reported.

## Probe synthesis

*Rink-amide linker attachment to aminomethyl polystyrene resin.* The Fmoc-Rink-amide linker (3 eq, 0.1 M) was dissolved in DMF

and Oxyma (3 eq) was added, and the mixture was stirred for 10 min. DIC (3 eq) was added and the mixture was stirred for further 1 min. The solution was added to the resin (200 mg, 0.7 mmol/g, 1 eq, pre-swollen in DCM) and shaken for 2 h. The resin was washed with DMF (3 × 10 mL), DCM (3 × 10 mL) and MeOH (3 × 10 mL). The coupling reaction was monitored by a Kaiser test.

*Fmoc deprotection.* To the resin (0.2 mmol, 1 eq, pre-swollen in DCM), 20% piperidine in DMF was added and the reaction mixture was shaken for 10 min. The solution was drained and the resin was washed with DMF (3 × 10 mL), DCM (3 × 10 mL) and MeOH (3 × 10 mL). This procedure was repeated twice.

*Amino acid couplings.* A solution of the *N*-Fmoc-protected amino acid, Fmoc-(EG)$_2$-OH (3 eq), or Fmoc-Lys(MR)-OH (1.7 eq) and Oxyma (3 eq or 1.7 eq) in DMF (0.07 M) was stirred for 10 min. DIC (3 eq or 1.7 eq) was added and the solution was stirred for further 1 min. The solution was added to the resin (1 eq, pre-swollen in DCM) and the reaction mixture was shaken for 30 min at 50 °C, except for Fmoc-Lys(MR)-OH that was shaken for 1 h at 50 °C. The solution was drained and the resin washed with DMF (3 × 10 mL), DCM (3 × 10 mL) and MeOH (3 × 10 mL). The coupling reactions were monitored by a Kaiser test.

*Cleavage and deprotection.* The resin (0.2 mmol, pre-swollen in DCM) was shaken 3 h in TFA/TIS/DCM (95/2.5/2.5, 5 mL). The solution was collected by filtration and the resin was washed with the cleavage cocktail. The combined filtrates were added to ice-cold ether and the precipitated solid was collected by centrifugation and washed repeatedly with cold ether (3 × 50 mL).

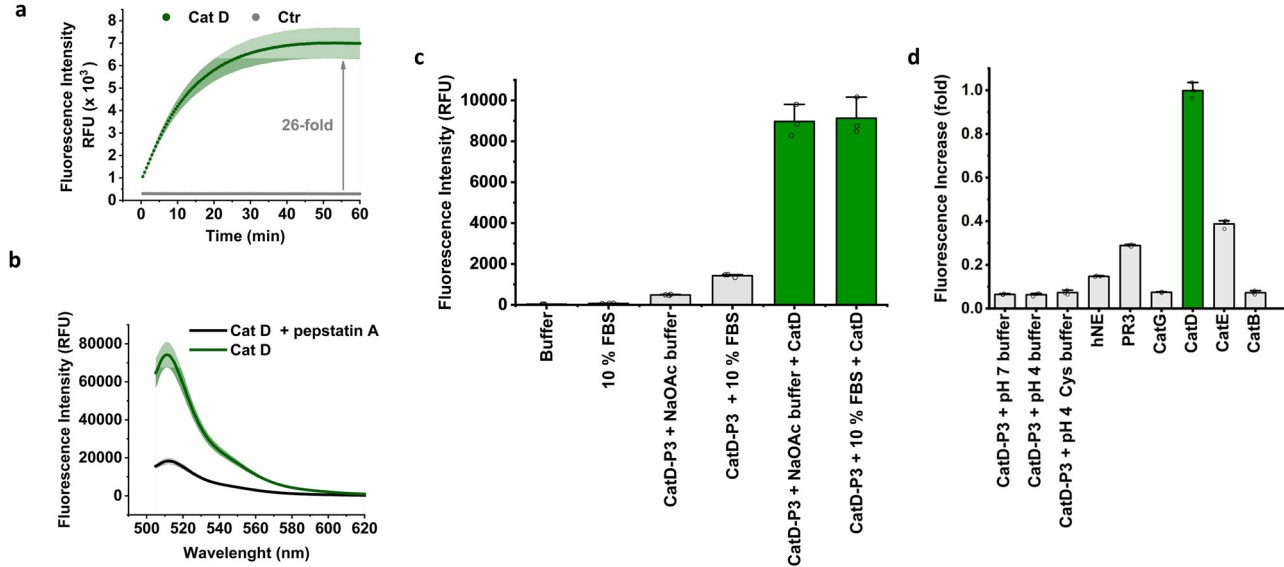

**Fig. 3 CatD-P3 activation in vitro. a** Time-dependent activation of **CatD-P3** (20 µM) in reaction buffer in the presence or absence of CatD (50 nM). Ctrl = probe alone. **b** CatD (50 nM) induced increase in the fluorescence intensity of **CatD-P3** (20 µM) with and without the inhibitor Pepstatin A (20 µM) ($n = 3$). **c** Activation of **CatD-P3** in reaction buffer and in pH 4 adjusted 10% FBS over 1 h. Fluorescence signal amplification was comparable under both conditions, with slightly higher background fluorescence in the 10% FBS media. **d** The graph represents the assessment of probe **CatD-P3** specificity for cathepsin D in comparison to other proteases associated with the immune response. **CatD-P3** (20 µM) was incubated with each protease (at a concentration of 5 nM) in their respective optimal reaction buffers for 1 h. To quantify the relative cleavage levels, the fluorescence increases were normalised to a maximum fold change of 1, which corresponds to the maximum cleavage achieved by cathepsin D. This normalisation was performed by dividing the fluorescence reading of each protease-activated probe by the maximum fluorescence value ($RFU_{max}$) obtained in the corresponding buffer for fully cleaved **CatD-P3** (20 µM), as detailed in the methods section. Measurements were taken in triplicate from distinct samples and mean values and standard deviation error bars were calculated and plotted using OriginLab.

*BODIPY-NHS attachment.* The crude peptide **1, 3** or **4** (20 mg, 1 eq, 10 mM), carrying a free amino group, was dissolved in anhydrous DMF, DIPEA (1 eq, 10 mM) was added, followed by the addition of BODIPY-NHS[30] **2** (1.5 eq, 15 mM). The reaction was stirred overnight at 40 °C under N₂. The solvent was removed *in vacuo* and the mixture washed with cold ether to remove excess/unreacted dye. And probe crudes were further purified by reverse phase semi-preparative HPLC (Supplementary Figs. 10–12).

*PEG-5K coupling.* Crude azide-peptide **5** (19 mg, 1.1 eq, 16 mM), was dissolved in anhydrous DMF and alkyne-PEG(5 K)-OMe (38 mg, 1 eq, 15 mM) was added, followed by a pre-mixed solution of CuI/THPTA (0.5/2.5 eq, final concentration 2 mM/10 mM). The reaction was stirred at 50 °C overnight under a N₂ atmosphere (the reaction was monitored by RP-HPLC). The solvent was evaporated *in vacuo* and purified by reverse phase semi-preparative HPLC (Supplementary Fig. 13)

*Probe purification.* All the probes were purified on a semi-preparative HPLC Agilent HP1100 system equipped with a reverse-phase column (Phenomenex Aeris 5 µm XB-C18 100 Å, 250 × 10 mm, 5 µm). The flow rate was 2 mL/min eluting with 0.1% HCOOH in H₂O (A) and 0.1% HCOOH in CH₃CN (B), with a gradient of 5 to 95% B over 25 min. Fractions containing the product were combined and the solvent removed via freeze-drying to give probes **CatD-P1, CatD-P2** and **CatD-P3** as orange solids in >95% purity. Full characterisation of the probes can be found in Supplementary Notes 1–4 and Supplementary Figs. 10–13.

**Biological evaluation.** For experiments using human neutrophil elastase, Cathepsin G or Proteinase 3 (purchased from Abcam or

Athens Research and Technology), the reaction buffer was 50 mM HEPES, pH 7.4, 0.75 NaCl, 0.05% Igepal CA-630 (v/v). Experiments using Cat D (Abcam) or E (BioLegend) used a buffer of 50 mM NaOAc, pH 4.0; while for Cathepsin B (Abcam), the reaction buffer was 50 mM NaOAc, pH 5.5, EDTA 1 mM, 4 mM Cys. Activity of the purchased proteases were validated using commercial substrates AAPV-pNA (for hNE and Proteinase 3), AAPF-pNA (for Cathepsin G), FR-pNA (Cathepsin B) following the manufacturers' protocols. Plate reader experiments were performed in a Biotek Synergy HT multi-mode reader ($\lambda_{ex}$ 485/20, $\lambda_{em}$ 520/20) using a 96-well plate (Life Technologies). Concentrations of probe stocks were calculated using a calibration curve (Supplementary Fig. 14).

*Time-dependent increase in fluorescence experiments.* The time-dependent increase in fluorescence was monitored over 1 or 2 h using a fluorescence microplate reader at 37 °C. **CatD-P3** (20 µM), 50 mM NaOAc, pH 4.0 buffer, enzymes (5 or 50 nM) and inhibitor Pepstatin A, where appropriate, were incubated in the wells for 30 min at 37 °C before adding the probe. Readings were taken immediately after addition of the probe and after every 30 s (the plate was shaken for 3 s before the start of the readings).

*Fluorometric experiments.* Probe **CatD-P3** (20 µM) was incubated with or without Cat D (50 nM) in presence of absence of pepstatin A (20 µM), in the corresponding reaction buffer for 1.5 h. The mixture (25 µL) was then diluted to a final volume of 150 µL in a quartz cuvette. Emission spectrum readings were recorded in a spectrofluorometric range from 500 to 700 nm, with an excitation wavelength of 490 nm.

*Specificity assays.* Cathepsins D, B or E (5 nM) was incubated with **CatD-P3** (20 µM) in the corresponding reaction buffer in a final

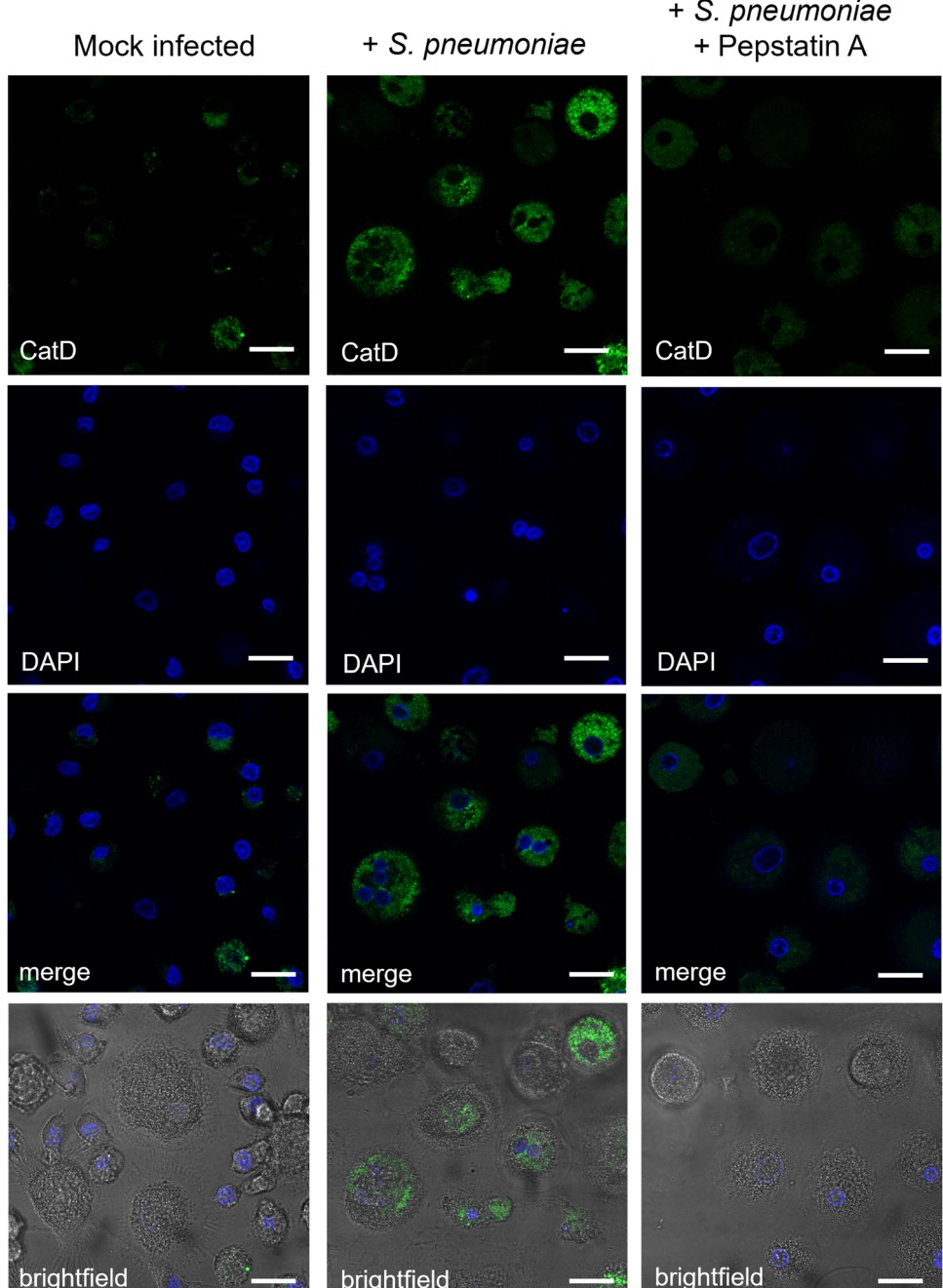

**Fig. 4 Detection of CatD activity in monocyte-derived macrophages.** Confocal fluorescence microscopy ($\lambda_{ex}$ 488, $\lambda_{em}$ 512 nm) images of the detection of CatD activity in monocyte-derived macrophages. Cells appear phenotypically normal, and the DAPI stained nuclei show no signs of apoptosis. CatD release from the lysosomes into the cell cytoplasm was observed in *S. pneumoniae* infected macrophages (central panels) challenged with *S. pneumoniae* followed by incubation with **CatD-P3** (10 μM). Mock infected macrophages (no bacteria) were used as a control (left panel). Co-incubation with the CatD inhibitor Pepstatin A (20 μM) (right panels) prevented **CatD-P3** activation resulting in reduced/no green fluorescence. Experimentally macrophages were challenged for 6 h with *S. pneumoniae* at a "Multiplicity of Infection" of 10:1, followed by incubation with **CatD-P3** (10 μM) for 4 h and $+/-$ Pepstatin A (20 μM). At 10 h the cells were washed, fixed in 2% paraformaldehyde and cell nuclei were counter-stained with DAPI ($\lambda_{ex}$ 405, $\lambda_{em}$ 461 nm). Scale bar $=$ 10 μm.

volume of 50 μL in a 96-well plate. For Cathepsin G, PR3 or hNE, the highly digestive enzyme Savinase (1 μL, 0.4 μ/mL) was used as a positive control (CatD is poorly active at pH 7). The time-dependent increase in fluorescence was monitored over 1 h using a fluorescence microplate reader at 37 °C. The fold increase in signal was calculated by dividing the fluorescence signal obtained by cleavage of each protease divided by the RFU of the fully cleaved probe by CatD (considered as RFU_max).

*Imaging of macrophages.* Human MDM were matured in 24-well Corning tissue culture plates over 14 days from Peripheral Blood Mononuclear cells obtained healthy volunteers by informed written consent, under ethical approval 21-EMREC-041. Following maturation, the MDM were removed from wells by treatment with Accutase (BioLegend) and re-seeded onto iBidi 8-well chamber slides. The cells were challenged with *S. pneumoniae* strain D39 (MOI 10, opsonised in human immune serum

for 30 min at 37 °C for a total of 10 h, with external bacteria being removed by washing after 4 h of initial phagocytosis[38]. After removal of external bacteria, **CatD-P3** (10 μM) was incubated with the cells for the remaining 6 h until the total bacterial incubation time of 10 h. In some experiments, the CatD inhibitor Pepstatin A was added to cell media at a concentration of 20 μM for the duration of the experiment. The MDM were then washed twice with PBS and fixed in 2% paraformaldehyde for 20 min at room temperature. Following fixation, cells were washed again with 3 × PBS, with DAPI (0.1 μg/mL) added in the 2nd PBS wash for 10 min. The slides were imaged on a Leica SP5 confocal microscope with a 63× objective, with **CatD-P3** imaged at 488 nm and DAPI imaged at 405 nm wavelength.

*Cytotoxicity assay.* Cytotoxicity of the **CatD-P3** probe was assessed using the CyQUANT LDH Cytotoxicity Assay Kit (Invitrogen), according to manufacturer's instructions. Briefly, **CatD-P3** (10 μM) was added to cultured MDMs in 24 well tissue culture plates for the desired timepoints. Supernatant was then removed, filtered through 0.22 μM syringe filters, and assayed for LDH activity. Untreated wells were used to assay Spontaneous LDH activity from cells, and cell lysis buffer was used to generate a Maximum LDH activity sample. % of cytotoxicity was calculated by the following formula: % Cytotoxicity = [(Compound LDH activity – Spontaneous LDH activity)/(Maximum LDH activity – Spontaneous activity)] x 100.

**Reporting summary**. Further information on research design is available in the Nature Portfolio Reporting Summary linked to this article.

## Data availability
Data supporting the paper is provided in the supplementary information section. The raw data for Fig. 3 has been provided as Supplementary Data 1.

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

## Acknowledgements

We would like to thank Engineering and Physical Sciences Research Council (EPSRC, UK) for Interdisciplinary Research Collaboration grants EP/R005257/1 and Medical Research Council SHIELD consortium grant MR/N02995X/1.

## Author contributions

M.R.-R. contributed with conceptualisation, synthesis and performed all the in vitro validation studies and manuscript writing. Z.L helped with the probe synthesis under the supervision of M.R.-R. B.J.M. performed the cell imaging and cytotoxicty assays. A.M.-F., A.L., D.D. and M.B. contributed with conceptualisation, data analysis and manuscript writing.

## Competing interests

The authors declare no competing interests.
