## [Peer review file · Communications Chemistry]

A fluorogenic, peptide-based probe for the detection of Cathepsin D in macrophagesReviewers' comments:

Reviewer #1 (Remarks to the Author):

The manuscript by Bradley and co-workers reports on the authors investigations into the development of a water soluble and pH independent, peptide based FRET probe for detection of the aspartic endo-protease, Cathepsin D. The pegylated probe (CatD-3) displayed good specificity for CatD over other proteases and was used to detect CatD activity in human, monocyte-derived macrophages that were challenged by exposure to bacteria. Moreover, incubation in the presence of a known inhibitor, Pepstatin A, resulted in a marked decrease in fluorescent signal. Overall the manuscript is well written and well-presented and the design of the FRET probe incorporating a PEG chain is novel. The finding that the probe is active across a broad pH range is significant and offers novelty over other recently reported FRET probes for CatD. The manuscript is suitable for publication in Communication Chemistry subject to minor corrections below.

Corrections

1. Line 135: 'of the' repeated
2. Could the authors comment on any steric factors that may diminish the effectiveness of the probe through incorporation of the large PEG chain i.e. does it in any way impair enzymatic activity?
3. Could the authors comment on the mechanism of cell uptake, is it via a passive mechanism?

Reviewer #2 (Remarks to the Author):

The manuscript shows a peptide probe that allows the monitoring of Cathepsin D activity both in solution and, in principle, in macrophages. The probe consists of a peptide sequence previously identified as a Cathepsin D substrate, to which a FRET pair has been added, resulting in signal activation when the enzyme is present.

Overall, the work is well presented and highly focused. However, the novelty of the work is not clear and fully justified. The use of a FRET pair to detect enzyme activity is not a novel concept. The authors report that the sequence used for the probe was previously described by Pimenta et al (*Biochimica et Biophysica Acta* 2001, 1544: 113-122) but in that work the authors already used a FRET pair (o-aminobenzoic acid and N-(2,4-dinitrophenyl)-ethylenediamine, blue region) to monitor Cathepsin D activity. Instead, in this case the authors choose a BODIPY derivative for their design, but other probes described in the literature, such as *ChemBioChem* 2022, 23, e202200319, use cyanine derivatives (green and red emission) for the detection of both Cathepsin B and Cathepsin D, and again in that paper the catalytic efficiency for Cathepsin D is higher than the value reported by the authors of this manuscript ($1.7 \text{ e}5 \text{ M}^{-1}\text{s}^{-1}$ vs $7.5 \text{ e}4 \text{ M}^{-1}\text{s}^{-1}$).

Compared to these two reports, the novelty seems to be the imaging of CatD activity in macrophages. In this case, it seems that in figure 4, panel a) right and panel b) left, the cells are treated under the same conditions, but the images are quite different. In panel a) right, the cells don't seem to be quite healthy, it would be good to add a brightfield image of the cells to see their morphology and to perform a cytotoxicity assay to check whether the probe is toxic to the cells or not. Also, in the methods section, the authors state "After removal of external bacteria, CatD-P3 (10 μM) was incubated with the cells for 10 h. The MDM were then 296 fixed in 2% paraformaldehyde for 20 min at room temperature..." are the cells washed before fixation? If not, is the peptide internalized into the cells before the cells are fixed or is it due to the fixation?

The authors state that the fluorescence of the probe is not affected by the pH, but there seem to be some differences between pH 4 and pH 9, are these statistically significant? Also, these experiments were performed by adding 10 μL of a 60 μM CatD-P3 solution in 50 mM NaOAc buffer pH 4 to 40 μL of buffers with pH 5.5, 7.4, and 9.2 to obtain the final 12 μM CatD-P3 solutions. Are the authors sure that the final pH of the solutions is 5.5, 7.4, 9.2? This is not obvious given the volumes, concentrations and the pKa of the buffers used.

Minor comments:

Figure 1 and 2 should be swapped for clarity.

C- and N-terminus/terminal should be italicized (N and C) both in the manuscript and in the SI.

In figure 3 d, CG should be labeled as the other Cathepsins (CatG) for clarity.

Figure S3 caption, the values of the λ_{ex} and λ_{em} are the same, the λ_{em} should be corrected.

Figure S6, instead of

in the figures, the authors should use [CatD-P3], which is the name of the probe in the manuscript.

Figure S8, it is not clear to me why the intensity of the peak at tR 0.5 changes between chromatograms when the length of the injection is changed.

Figure S10 caption, rt 4.6 min should be changed to tR 4.6 min.

Reviewer #3 (Remarks to the Author):

This is a pleasingly concise description of a new reported probe for cathepsin D, a protease of biomedical interest in understanding antibacterial activity of macrophages. From a technical perspective the work is good, though a few things need clarifying.

Specific points

L86 – I think there needs to be a bit more on the selectivity of cleavage – it's not clear how the reported selectivity connects with the Phe-Phe cleavage....

L137 – be a bit more specific re cellular compartment – pH not the only issue?

L133 – how do the kinetic parameters compare to simple peptide substrates?

L160 – Add reference to macrophage/bacteria/Cat D assay

L181 – 'embodying' or 'enabling'?

Fig S4 – show relevant MALDI MS data for specific cleavage if possible.

If possible, give NMR data for the probes. What is the evidence for maintenance of stereochemistry?

Figure 4 – can repeats be added to the SI?

Reviewer #1 : The manuscript is suitable for publication in Communication Chemistry subject to minor corrections below.

Corrections

1. Line 135: 'of the' repeated

The typo has been corrected in the main manuscript.

2. Could the authors comment on any steric factors that may diminish the effectiveness of the probe through incorporation of the large PEG chain i.e. does it in any way impair enzymatic activity?

This is an interesting point. The 5000 Da PEG unit is distal to the cleavage site (so it should not interfere), but it is also quite large (and will be well solvated) – so quite some steric bulk. However, if there were steric hindrance impeding enzymatic activity, we would expect slower activation of the CatD-P3 (the probe with the long PEG chain) compared to CatD-P2 (the much short PEG unit). This was not observed (see Fig S3). In addition, the kinetic parameters obtained for CatD-P3 are comparable to those reported for other linear peptides. Literature examples include the substrates by Pimenta⁸ or Yonezawa⁹ with catalytic efficiencies of $k_{cat}/K_m = 6 \times 10^5 \text{ M}^{-1} \text{ s}^{-1}$ and $k_{cat}/K_m = 6.2 \times 10^5 \text{ M}^{-1} \text{ s}^{-1}$. The substrates reported by Zhang¹¹ (Zhengxiao) had kinetic parameters of $k_{cat}/K_m = 1.7 \times 10^5 \text{ M}^{-1} \text{ s}^{-1}$. Likewise, these are similar to our probe ($7.5 \times 10^4 \text{ M}^{-1} \text{ s}^{-1}$). These values of k_{cat}/K_m are similar to those measured for FRET substrates for other proteases.⁷ Commercial fluorogenic substrates have substantially slower kinetics (Enzo, BML-P145-0001, $k_{cat}/K_m = 1.09 \times 10 \text{ M}^{-1} \text{ s}^{-1}$) and generally require high levels of DMSO, due to their poor solubility.

We have added the sentence *“The presence of the large PEG unit had little effect on the enzymatic activity of **CatD-P3**, with a good binding affinity (K_M of 8 μM) and reasonable turnover number (k_{cat} of 0.6 s^{-1}).*

3. Could the authors comment on the mechanism of cell uptake, is it via a passive mechanism?

Macrophages are immune cells that are phagocytically highly active. We believe that the main mechanism of cell uptake is phagocytosis.

We have added to the text: “Since macrophages are professional phagocytes, we believe that probe uptake is via a combination of passive diffusion and phagocytosis.”

Reviewer #2:

Overall, the work is well presented and highly focused. However, the novelty of the work is not clear and fully justified. The use of a FRET pair to detect enzyme activity is not a novel concept. The authors report that the sequence used for the probe was previously described by Pimenta et al (Biochimica et Biophysica Acta 2001, 1544: 113-122) but in that work the authors already used a FRET pair (o-aminobenzoic acid and N-(2,4,dinitrophenyl)-ethylenediamine, blue region) to monitor Cathepsin D activity. Instead, in this case the authors choose a BODIPY derivative for their design, but other probes described in the literature, such as ChemBioChem 2022, 23, e202200319, use cyanine derivatives (green and red emission) for the detection of both Cathepsin B and Cathepsin D, and again in that paper the catalytic efficiency for Cathepsin D is higher than the value reported by the authors of this manuscript ($1.7 \times 10^5 \text{ M}^{-1}\text{s}^{-1}$ vs $7.5 \times 10^4 \text{ M}^{-1}\text{s}^{-1}$).

The FRET pair chosen by Pimenta⁸ (o-aminobenzoic acid and N-(2,4,dinitrophenyl) works in the UV/blue region and allows evaluation in vitro, but is incompatible with cell imaging. The reviewer is correct in that other probes have been described in the literature such as those published by Zhang who used cyanine derivatives.¹¹ We mentioned this paper in our manuscript "Cy3/5-based FRET substrate for Cathepsin D has been reported,¹¹ but this was conjugated to a 20-mer oligonucleotide."

The attachment to a 20-mer oligonucleotide is far from ideal (although it makes the probe soluble), but it makes scale-up challenging. Thus, in their paper the authors work in the nM range of substrates which makes determination of K_m values challenging (usually you need to work between $0.5-2 \times K_m$) with an enzyme to substrate ratio of 1:20 as standard in their assays (this is far from ideal – but a consequence of using an oligonucleotide as a tag). The Lineweaver Burke plots in the reported ChemBioChem paper also cross the +ve x-axis (which would give a -ve K_m value). The probe in this paper also needed to be transfected to get into cells due to the oligonucleotide tag).

We chose the BODIPY/Methyl Red FRET pair as this allows imaging in live cells, has a bright, green emitting fluorophore and a highly efficient quencher – and gave a probe that is highly specific, fully soluble and can be produced readily by SPSS.

Compared to these two reports, the novelty seems to be the imaging of CatD activity in macrophages. In this case, it seems that in figure 4, panel a) right and panel b) left, the cells are treated under the same conditions, but the images are quite different. In panel a) right, the cells don't seem to be quite healthy.

It would be good to add a brightfield image of the cells to see their **morphology** (done - see below) **and to perform a cytotoxicity assay** to check whether the probe is toxic to the cells or not (done - see below)

Also, in the methods section, the authors state "After removal of external bacteria, CatD-P3 ($10 \mu\text{M}$) was incubated with the cells for 10 h. The MDM were then 296 fixed in 2% paraformaldehyde for 20 min at room temperature..." **are the cells washed before fixation?** (yes the cells were washed before fixation - see below). If not, is the peptide internalized into the cells before the cells are fixed or is it due to the fixation?

The cytotoxic effect of our probe was evaluated by performing a **LDH cytotoxicity assay**. The additional text has been added to the manuscript:

“LDH cytotoxicity assays on macrophages incubated with 10 μ M CatD-P3 showed low levels of toxicity at 1 hour (0.27% \pm 0.2%) and 6 hours (6.3% \pm 0.68%)(Supplementary Information Figure S7). Additionally, the cells appeared phenotypically normal after probe incubation by phase contrast (Fig 4, lower panels).”

And the following figure has been added to the supplementary information.

Figure S7. Cytotoxicity of CatD-P3 probe on monocyte derived macrophages (MDM), assessed by LDH release. After incubation with 10 μ M CatD-P3 probe, supernatant from MDMs was monitored at 1 and 6 hours, and compared to maximum LDH release from lysed cells. LDH release is shown as a percentage of maximum LDH release detected by this assay, \pm SEM. N = 3 assays.

Brightfield images have been added to Figure 4. Extensive work has been reported previously on macrophage bacterial infection models and the same previously characterised infection model was used here ¹⁰ and the results are consistent with those obtained at the 10-hour timepoint. In the images, cells appear phenotypically normal, and DAPI stained nuclei show no signs of apoptosis.

Figure 4. Confocal fluorescence microscopy (λ_{ex} 488, λ_{em} 512 nm) images of detection of CatD activity in monocyte-derived macrophages. CatD release from lysosomes into the cell cytoplasm was induced by challenging the macrophages for 10 hours in the presence of *S. pneumoniae* at a Multiplicity of Infection of 10:1, with incubation with **CatD-P3** (10 μ M) with **(A)** or without **(B)** CatD inhibitor Pepstatin A (20 μ M) from 4 hours onwards after removal of extracellular bacteria by washing. Mock infected macrophages were used as a control. At 10 hours the cells were washed, fixed in 2% paraformaldehyde and cell nuclei were counter-stained with DAPI (λ_{ex} 405, λ_{em} 461 nm). Scale bar = 10 μ m.

We confirm that the probe was **internalised prior to fixation** - Cells were washed twice before fixation as standard and this has been clarified in the legend to **Figure 4**.

The authors state that the fluorescence of the probe is not affected by the pH, but there seem to be some differences between pH 4 and pH 9, are these statistically significant? Also, these experiments were performed by adding 10 μ L of a 60 μ M CatD-P3 solution in 50 mM NaOAc buffer pH 4 to 40 μ L of buffers with pH 5.5, 7.4, and 9.2 to obtain the final 12 μ M CatD-P3 solutions. Are the authors sure that the final pH of the solutions is 5.5, 7.4, 9.2? This is not obvious given the volumes, concentrations and the pKa of the buffers used.

Thank you for the comment. The effect of the $\frac{1}{4}$ dilution into a different buffer was indeed underestimated in Figure S5. Following the reviewer comments, we have assessed the effect of the dilution in the different buffers. Although we found no significant effect for buffers going from pH 4.0 to pH 5.5 and 7.4, the pH 9.2 (5 mM borate buffer) did show acidification upon addition of the pH 4 buffer (pictures attached below). We have therefore removed the pH 9.2 data as this is not relevant in a cellular context (the enzyme works at pH 4.0). Statistical analysis has been included in the legend and shows no significant differences were observed when comparing the fluorescence signals of cleaved and uncleaved at pH 4.0 with the corresponding signals at pH 5.5 and 7.4.

Figure S5. The effect of pH on the fluorescence signal. **CatD-P3** (60 μ M) was incubated 1.5 h in the reaction buffer (50 mM NaOAc pH 4.0) in a final volume of 200 μ L with or without CatD (50 nM, Athens Biotechnology). The solution was then diluted in the corresponding buffer (pH buffers (50 mM NaOAc pH 4.0 or 5.5, PBS pH 7.4) to give final concentrations in the well of 12 μ M (50 μ L volume) and fluorescent intensities were measured using a fluorescence microplate reader (Biotek Synergy HT multi-mode reader) ($\lambda_{ex/em}$ 485/20, emission 520/20) at 37 $^{\circ}$ C. Two tailed paired Student's test statistical analysis showed no significant differences (p-values > 0.05) in data comparisons where asterisk (*) refers to values compared to uncleaved probe at pH 4 and dash (#) refers to values compared to fully cleaved probe at pH 4.

BODIPY dyes are known for their pH insensitivity.^{1,2} This is also demonstrated by the specificity assays with different protease being run in different buffers (of different pHs) giving consistent backgrounds and maximum fluorescence levels for the uncleaved and the fully cleaved probe.

Minor comments:

Figure 1 and 2 should be swapped for clarity. We agree - the figure order has been swapped.

C- and N-terminus/terminal should be italicized (N and C) both in the manuscript and in the SI. Format has been updated.

#In figure 3 d, CG should be labelled as the other Cathepsins (CatG) for clarity. Has been updated

Figure S3 caption, the values of the lambda ex and lambda em are the same, the lambda em should be corrected. Has been corrected

Figure S6, instead of in the figures, the authors should use [CatD-P3], which is the name of the probe in the manuscript. The axis title has been updated.

Figure S8, it is not clear to me why the intensity of the peak at tR 0.5 changes between chromatograms when the length of the injection is changed. Injection time was the same, but running time was different. To simplify and improve clarity we have removed one of the traces and just report the 15 minutes run.

Figure S10 caption, rt 4.6 min should be changed to tR 4.6 min – updated.

Reviewer #3:

This is a pleasingly concise description of a new reported probe for cathepsin D, a protease of biomedical interest in understanding antibacterial activity of macrophages. From a technical perspective the work is good, though a few things need clarifying.

Specific points

L86 – I think there needs to be a bit more on the selectivity of cleavage – it's not clear how the reported selectivity connects with the Phe-Phe cleavage....

Several studies have been carried out to understand substrate specificity of CatD. Majer³ (Majer, 1997) used modified pepstatin inhibitors and modified several positions (P4, P3, P1, P2', and P3'). Scarborough⁴ used synthetic FRET substrates to study the preference of the enzyme for its S2 and S3 pockets by modifying the P2 and P3 positions of the substrate. Beyer⁵ and Arnold⁶ studied the prime side subsite specificity by modification in the P2' and P3' using synthetic substrates and assessing degradation of natural substrates at positions P1-P4', respectively. Pimenta⁸ performed a systematic study using a library of modified substrates derived from kallistatin. These results found that hydrophobic residues were preferred at P1 P1', with Phe, Leu or Met combinations showing the best cleavages and P1' having a more influence in cleavage over P1. The P2 site preferred Ala, Leu or Glu, and this site was a key modification within the kallistatin library study (modification of this residue lead to the best substrate). The P3 and P3' sites also prefer hydrophobic residues. Interestingly, the P2' site showed different results in different studies, with Beyer reporting preference for positively charged amino acids while Pimenta showed preference of Ser; however, in both studies, various other residues were tolerated at this position (for instance Asp and Ala showed similar kinetic parameters), a finding that was reported by other authors.⁸ The S2 pocket is a hydrophobic pocket, but long chain charged residues can "stick out" of the pocket and be exposed to solvent.

We have added the text to the manuscript: *"When choosing a substrate to build an optical probe to be used in cells and potentially in vivo, control of cross-selectivities is vital and it is important to design a specific substrate that is not cleaved by related proteases. One of the most abundant and active proteases in the lysosome is Cathepsin B, which prefers substrates bearing Phe-Arg or Arg-Arg sequences at the P2-P1 sites. Previously reported, highly efficient cleavable substrates for CatD, have the sequence Phe-Arg (P1'-P2') which would also be cleaved by Cathepsin B. The optimal substrate reported by Pimenta was chosen for its cleavage site combination, avoiding the Phe-Arg and Arg-Arg combination, and targeting its high catalytic efficiency for the substrate."*

L137 – be a bit more specific re cellular compartment – pH not the only issue?

We have rewritten: *"CatD has an optimal pH of 4.0, a pH found in the lysosomes and at this pH the cleaved probe is highly fluorescent. Thanks to its pH insensitivity (see Figure S5), during transport to or from the phagosome, for example when leaking into the cytoplasm (see Figure 4), the cleaved probe will remain fluorescent."*

We believe that the experiments showing activation in serum (showing a similar activation profile as those in vitro), the stability of the background fluorescence in these complex environments and the repeated imaging studies support the statements on robustness and photostability. We have added a

figure to the supporting information section where the time dependent activation and background is fully reported (Fig S9).

We have updated the text: “*CatD-P3* was photostable and functional in serum i.e., in an environment with an abundance of proteins and other metabolites (Figure 3c and SI Figure S9).”

Figure S9. Time dependent activation of **CatD-P3** in reaction buffer and at pH 4.0 adjusted to 10% FBS.

L133 – how do the kinetic parameters compare to simple peptide substrates? Please see our response to reviewer 1.

L160 – Add reference to macrophage/bacteria/Cat D assay. A reference has been added to the sentence.⁹ Bewley *et al.*, 2021 (reference 3 in manuscript).

L181 – ‘embodying’ or ‘enabling’? Thank you for the suggestion, that has now been updated.

Fig S4 – Show relevant MALDI MS data for specific cleavage if possible. The supplementary information has a figure showing the specific cleavage and the masses of the two fragments of the probe. **Fig S4.**

If possible, give NMR data for the probes. Peptides were characterised HRMS, MALDI-TOF MS, LC-MS and analytical HPLC which is usual for such peptides.

What is the evidence for maintenance of stereochemistry?

The peptides were assembled by solid phase peptide synthesis (SPPS) using chiral Fmoc-protected amino acids. The coupling methods used minimise any epimerisation (DIC/Oxyma) and any epimerisation would have led to diastereomers that would appear as additional peaks on the analytical HPLC and separated during purification.

Figure 4 – can repeats be added to the SI? We have included this in the supplementary information section.

Figure S8. Additional confocal fluorescence microscopy examples of detection of CatD activity in monocyte-derived macrophages. Conditions as per Figure 4 in the main text - incubation with CatD-P3 (10 μ M) +/- *S. pneumoniae* (MOI 10) and +/- CatD inhibitor Pepstatin A (20 μ M). Cell nuclei were counter-stained with DAPI. Scale bar = 10 μ m.

References

1. A Loudet and K Burgess, *Chemical Reviews* 2007 107 (11), 4891-4932 // Murale, D.P., Haque, M.M., Hong, K.T. and Lee, J.-S. (2021), A Pyridinyl-Pyrazole BODIPY as Lipid Droplets Probe. *Bull. Korean Chem. Soc.*, 42: 111-114.
2. S. Zhang, T. Wu, J. Fan, Z Li, N Jiang, J Wang, B Dou, S Sun, F Song and X Peng *Org. Biomol. Chem.*, 2013,11, 555-558
3. P. Majer, J. R. Collins, S. V. Gulnik and J. W. Erickson, *Prot Sci*, 1997, 6, 1458-1466. 178.
4. P. E. Scarborough and B. M. Dunn, *Protein Engineering, Design and Selection*, 1994, 7, 495-502. 179.
5. B. M. Beyer and B. M. Dunn, *Prot Sci*, 1998, 7, 88-95. 180.
6. D. Arnold, W. Keilholz, H. Schild, T. Dumrese, S. Stevanović and H. G. Rammensee, *Eur J Biochem*, 1997, 249, 171-179
7. S. Mittoo, L.E. Sundstrom, M. Bradley, *Anal Biochem*, 2003, 319, 234-238.
8. D. C. Pimenta, A. Oliveira, M. A. Juliano and L. Juliano, *BBA Protein Struct Mol Enzym*, 2001, 1544, 113-122
9. Hiroo YONEZAWA et al. *Bioscience, Biotechnology, and Biochemistry*, Volume 63, Issue 8, 1 January 1999, Pages 1471–1474.
10. Bewley MA, Marriott HM, Tulone C, Francis SE, Mitchell TJ, Read RC, et al. *PLoS Pathogens* 2011, 7: e1001262
11. Zhang Z, Nakata E, Shibano Y & Morii T. FRET-Based Cathepsin Probes for Simultaneous Detection of Cathepsin B and D Activities. *ChemBioChem* 2022, 23: e202200319.

REVIEWERS' COMMENTS:

Reviewer #1 (Remarks to the Author):

I confirm that all of my comments have been fully addressed by the authors, I recommend the manuscript for publication.

Reviewer #2 (Remarks to the Author):

The authors have improved the manuscript, but there are still a couple of things that need to be revised before acceptance:

1. The novelty of the work should also be highlighted in the manuscript, for example by adding the limitations of the two examples cited: Pimenta et al. incompatibility with cell imaging; Zang et al. need to use transfection agents to internalise the probe.

2. The description of figure 4, both in the main text and in the legend, does not clarify why the images of the second and third columns show a different green fluorescence pattern when, theoretically, according to the legends of the figure, cells were treated under the same conditions. One can assume that the second column represents cells with lysosomal accumulation of the probe and the third column represents cells with cytoplasmic leakage of the probe, but this should be clarified in the main text and the legend and by not including the third column in the section B of the figure. Also, the figure 4 legend seems to be incorrect, "... at a Multiplicity of Infection of 10:1, with incubation with CatD-P3 (10 uM) with (A) or without (B) CatD inhibitor Pepstatin A..." according to the images and the main text, panel A is the treatment without and panel B is the treatment with Pepstatin A.

Reviewer #3 (Remarks to the Author):

I'm very happy to recommend publication of this nice manuscript – the authors have addressed all my comments and those of the other reviewers more than adequately.

Following the new comments from reviewer 2, we have modified/added two statements in the manuscript as requested (see below).

The authors have improved the manuscript, but there are still a couple of things that need to be revised before acceptance:

1. The novelty of the work should also be highlighted in the manuscript, for example by adding the limitations of the two examples cited: Pimenta et al. incompatibility with cell imaging; Zang et al. need to use transfection agents to internalise the probe.

- To highlight the disadvantages of the previously reported probes, lines 70-73 now read “but this was conjugated to a 20-mer oligonucleotide and used a substrate that showed poor cleavage kinetics. In addition, DNA oligomer attachment to a protease substrate poses a scale-up challenge and requires cell transfection for intracellular delivery.”
- Lines 85-87 now read “...but adapted to be compatible with live cell imaging by improving solubility and moving the emission of the fluorescence into the green region of the spectrum”. This highlights the optimisation of the probe from Pimenta’s original design.

2. The description of figure 4, both in the main text and in the legend, does not clarify why the images of the second and third columns show a different green fluorescence pattern when, theoretically, according to the legends of the figure, cells were treated under the same conditions. One can assume that the second column represents cells with lysosomal accumulation of the probe and the third column represents cells with cytoplasmic leakage of the probe, but this should be clarified in the main text and the legend and by not including the third column in the section B of the figure. Also, the figure 4 legend seems to be incorrect, “.... at a Multiplicity of Infection of 10:1, with incubation with CatD-P3 (10 uM) with (A) or without (B) CatD inhibitor Pepstatin A...” according to the images and the main text, panel A is the treatment without and panel B is the treatment with Pepstatin A.

We agree that the legend was slightly unclear and the interpretation of the two experiments side-by-side was slightly confusing. In light of reviewer 2 comments, we have decided to switch this figure with a figure from the Supplementary figure S8 (and have rewritten the figure legend (both are shown below).

Figure 4. Confocal fluorescence microscopy (λ_{ex} 488, λ_{em} 512 nm) images of the detection of CatD activity in monocyte-derived macrophages. Cells appear phenotypically normal, and the DAPI stained nuclei show no signs of apoptosis. CatD release from the lysosomes into the cell cytoplasm was observed in *S. pneumoniae* infected macrophages (central panels) challenged with *S. pneumoniae* followed by incubation with **CatD-P3** (10 μM). Mock infected macrophage (no bacteria) were used as a control (left panel). Co-incubation with the CatD inhibitor Pepstatin A (20 μM) (right panels) prevented **CatD-P3** activation resulting in reduced/no green fluorescence. Experimentally macrophages were challenged for 6 hours with *S. pneumoniae* at a Multiplicity of Infection of 10:1, followed by incubation with **CatD-P3** (10 μM) for 4 hours and +/- Pepstatin A (20 μM). At 10 hours the cells were washed, fixed in 2% paraformaldehyde and cell nuclei were counter-stained with DAPI (λ_{ex} 405, λ_{em} 461 nm). Scale bar = 10 μm .

The supplementary Figure S8 is now as follows (figure legend has also been modified):

Supplementary Figure 8. Confocal fluorescence microscopy (λ_{ex} 488, λ_{em} 512 nm) images showing the detection of CatD activity in monocyte-derived macrophages challenged with bacteria. **(A)** Mock infected macrophages (no bacteria) showing negligible levels of fluorescence (in the presence of **CatD-P3** (10 μM)). **(B)** showing CatD release from lysosomes into the cell cytoplasm induced by challenging macrophages with *S. pneumoniae* in the presence of **CatD-P3** (10 μM). **(C)** is a replicate of the conditions in **(B)** and **(D)** shows how the addition of the CatD inhibitor Pepstatin A (20 μM) prevented activation of CatD-P3 in infected macrophages.

In these experiments, macrophages were incubated for 6 hours in the presence or absence (Mock) of *S. pneumoniae* (at a Multiplicity of Infection of 10:1), while in the presence of **CatD-P3** (10 μM) Pepstatin A (20 μM) was added after 4 hours. At 10 hours the cells were washed, fixed in 2% paraformaldehyde and cell nuclei were counter-stained with DAPI (λ_{ex} 405, λ_{em} 461 nm). Scale bar = 10 μm . In the images, cells appear phenotypically normal, and the DAPI stained nuclei show no signs of apoptosis.